# Photonic Signal Processing in Phase-Coded Lidar System

**Shuyu Chen [1], Long Wu [2,\*], Lu Xu [3,\*], Yong Zhang [4] and Jianlong Zhang [4]**

1 Keyi College of Zhejiang Sci-Tech University, Shaoxing 312369, China; chenshuyu@ky.zstu.edu.cn
2 School of Computer Science and Technology, Zhejiang Sci-Tech University, Hangzhou 310018, China
3 School of Information Science and Technology, Zhejiang Sci-Tech University, Hangzhou 310018, China
4 Institute of Optical Target Simulation and Test Technology, Harbin Institute of Technology, Harbin 150001, China; zzyyyy@hit.edu.cn (Y.Z.)
\* Correspondence: wulong@zstu.edu.cn (L.W.); xulu@zstu.edu.cn (L.X.)

**Abstract:** The next generation of lidar systems needs to adapt to variable environments with broadened bandwidth for increased resolution. Due to their digital components, conventional lidar systems, especially imaging lidar systems, suffer from limited detector bandwidth and sampling frequency. However, photonics devices can provide a reliable technical solution with high precision and ultrabroad bandwidth. This paper presents a photonic signal processing structure for a phase-coded lidar system. Two acousto-optic modulators (AOMs) are adopted in the proposed architecture. One is used for phase-coded laser signal modulation, and the other is used for demodulation. The echo laser signal is directed to the AOM performing demodulation before the sampling of the detector, accomplishing the multiplication of the echo laser signal and the electric reference signal. The detector is controlled to accumulate the demodulated laser signal. The AOM and detector transfer the correlation calculation from electrical signals processing to photonic signals processing. This photonics-based structure greatly decreases the sampling frequency of the detector without extending the width of the laser pulses, which achieves high resolution with low sampling speed. Photonic signal processing has the promising potential of simultaneously processing signals of multiple pixels. It is going to be an effective solution for imaging lidar systems to increase resolution with available low-cost devices.

**Keywords:** lidar; photonic signal processing; phase-coded pulse compression

## 1. Introduction

In the next generation of radar systems, new developing technologies are required for target detection at long range and with increased resolution in variable environments with broadened bandwidth. The lidar system, especially the imaging lidar system [1–3], also demands a revolution in signal processing, with a much wider bandwidth. With the improvement of MEMS scanners [4,5], the pulsed lidar systems, such as phase-coded lidar [6,7], use narrow pulses to achieve better resolution. However, suffering from the limited bandwidth and sampling frequency of the digital components, the performance of lidar systems is restricted. In contrast, photonics can provide high precision [8], ultrabroad bandwidth [9] and low phase errors [10], with the advantages of both the flexible generation [11,12] and detection of extremely stable signals [13] and their precise direct digitization without down conversion [14,15]. In 2015, Pérez first built a software-defined reconfigurable microwave photonics processor which was capable of performing all the main functionalities by the suitable programming of its control signals [16]. In 2018, Wang built an all-optical phosphorene phase modulator with enhanced stability under ambient conditions [17]. All these efforts have explored the possibilities of replacing electrical devices with photonic devices. Most notably, the group of Yao has made great progress in photonic signal processing. In 2015, they proposed a technique to achieve broadband and precise microwave time reversal using a single linearly chirped fiber Bragg grating [18].

From 2016 to 2022, the group reported the microwave signal convolution calculation based on microwave photonic signal processing technologies [19–22], which greatly improve the signal processing bandwidth. These photonic signal processing techniques have been already applied in large-scale imaging lidar since 2020 [23,24]. Because photonic signal processing is capable of providing higher bandwidth and lower signal interference, the application of photonic signal processing in lidar systems is promising.

The approach of time of flight (ToF) measurement is considered to be a promising technique [25]. The approach known as direct ToF (dToF) calculates the time directly from an accurate time base [26,27]. It is generally considered to be a low-resolution technology, requiring complex (and expensive) mechanical scanning to achieve high resolutions. The approach of indirect ToF (iToF) calculates distance based on a phase shift to a known reference signal. This technique suits high-resolution applications with less detection range [28]. However, the iToF technique may extend the ranging depth to avoid the ambiguity of phase by combining the detection of double frequencies. The approach makes iToF very useful in forward-looking applications where other technologies have limitations.

The phase-coded lidar system has great advantages in long-distance measurement with high resolution [29]. Yang built a phase-coded modulation lidar based on a ghost imaging algorithm to get 3D images of distant targets in 2020 [30]. Ding presented a coded-pulse-bunch-laser-based single-photon lidar for fast long-distance ranging in 2022 [31]. In this scenario, the collection of every returned laser pulse is required and expensive detectors with broad bandwidth and expensive electrical signal processors are used for complicated calculations. For instance, the resolution of a phase-coded lidar system is limited to 15 m when the bit width is 100 ns and the sampling frequency is 10 MHz. A PIN detector can easily achieve such high frequency, whereas an ICCD will never do so. Lidar systems with low sampling frequency, such as imaging array lidar systems, are confronted with great difficulty regarding the conflicts of high resolution and low detector bandwidth [32,33].

In this paper, the structure of photonic signal processing in the phase-coded lidar system is proposed to solve the conflict between the range resolution and sampling frequency of detectors. This photonics-based structure decomposes the correlation calculation into multiplication and addition steps achieved by the demodulator and detector. The achieved range resolution is the same as the sub-pulse width of the modulated sequence. Additionally, the accuracy has been greatly improved. The experiment has shown that the system will achieve a 15 m range resolution with a sampling period of 1.6 μs. The error of the measurement reached 0.5 m. The photonic signal processing method has transferred the electric signal processing to optical signal processing. Moreover, as the interference of laser beams is little enough, the photonics devices are capable of processing parallel optical signals, which may greatly improve signal bandwidth. Therefore, the proposed architecture has great implementation potentialities in imaging lidar systems.

## 2. Photonic Signal Processing in Lidar Systems

Generally, the detectors in conventional lidar systems collect all laser signals and transfer the laser signals to electrical signals, which are then applied to the signal processing module. Although the mature signal processing algorithm and electronic circuit design greatly decrease the difficulty of lidar system design, the performance of the system is restricted by the sampling frequency of the detectors. Most notably, the imaging lidar systems with array detectors are not capable of recovering broadband signals.

Because the photonics devices are capable of providing high precision and broad bandwidth, the proposed lidar system transfers the signal processing from electrics to photonics, as shown in Figure 1.

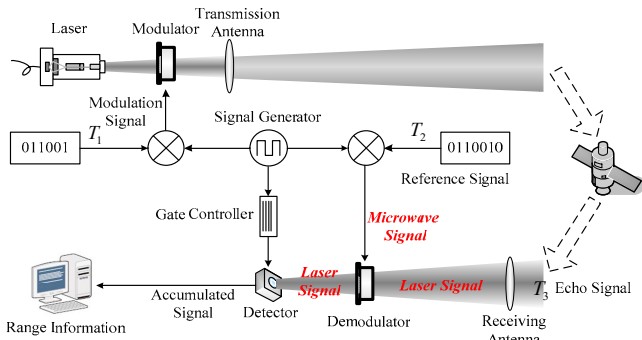

**Figure 1.** The photonics-based phase-coded lidar system.

In the system, an AOM is used to generate the phase-coded laser sequence at the transmission arm. Another AOM, as the demodulator, is placed before the detector to control the passage status of the echo laser pulses according to the electric reference sequence, achieving the multiplication of the echo and reference sequences. The detector integrates the laser signals during the demodulation period and outputs the values of the correlation calculation of the echo laser sequence and the electric reference sequence at a very low speed.

In the transmission arm, the generated electrical modulation pulse signals with pulse width $\tau$ are sent to the modulator. The laser with the constant intensity is modulated by the modulator into a phase-coded sequence, $T_1$, and transmits through the transmission antenna illuminating the target. The generated reference sequence $T_2$, which is an electrical signal, is sent to the demodulator and controls the passing status of the echo laser sequence $T_3$ in the demodulator, which is collected by the receiving antenna. The detector collects the laser signals which transmit through the demodulator and accumulates all the energy of the signals during the opening period, which is controlled by the gate controller. The output sequence of the detector is used to calculate the range information of the target in the computer by counting the position of the highest signal intensity.

In this lidar system, the correlation calculation of the echo and reference sequences does not emerge in the computer in the form of electrical signals. The correlation calculation is decomposed into three parts. (1) Multiplication operation: The demodulator is controlled by the electrical reference sequence $T_2$ to decide the passage status of the echo laser pulses. This photonics instrument achieves the multiplication of the echo sequence in the form of laser signals and the reference sequence in the form of electrical signals. Typically, the demodulator can be an acousto-optic modulator (AOM) or an electro-optic modulator (EOM). (2) Addition operation: The detector conducts the addition of multiplied sequences by the accumulation of laser signals under the condition of a controlled opening gate. Because the accumulation period equals the period of the reference sequence, the output speed of the detector is quite low, which is fit for the defect of the array detectors. (3) Shift operation: The reference sequence is one bit more than the transmitted sequence, which will automatically shift the transmitted sequence by one bit. The computer recognizes the correlation peak position within the output of the detector and calculates the phase shift. With these three steps, the conventional electrical correlation calculation is transformed to the optical correlation calculation with the collaboration of photonics-based devices.

As shown in Figure 2, a demonstration experiment was proposed to validate the proposed photonics-based phase-coded lidar system.

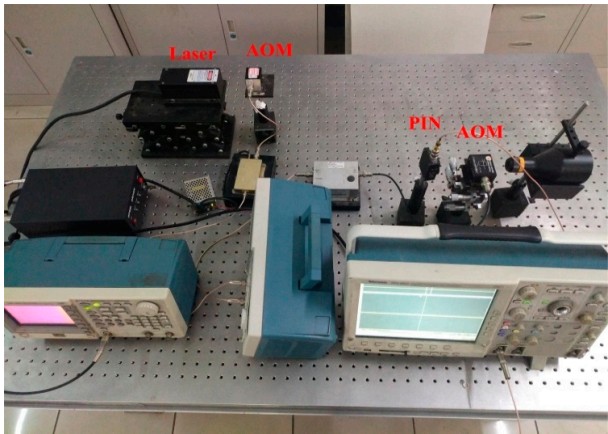

**Figure 2.** The demonstration experimental setup.

The system employed a CW laser transmitter emitting a laser signal with constant intensity, at the 532 nm wavelength, with the output power 75 mW. The laser was modulated to the coded pulse sequence by the acousto-optic modulator (AOM RF200 of Qingjin) (Shenzhen, China). The signal generator (AFG3102C of Tektronix) (Beaverton, OR, USA) generated two synchronous signals, the modulation sequence and reference sequence. The modulation sequence was selected as an m-sequence of 15 bits long with a bit width of 100 ns. Therefore, the periods of the modulation and reference sequences were 1.5 µs with 15 bits and 1.6 µs with 16 bits. To simplify the experiment, the activation period and silence period were both set to 24 µs.

Due to the Prague diffraction effect of the AOM, the direction of the modulated laser signal was deflected to the first-order diffraction fringe. Because the intensity of the laser signal was not high enough to support remote target detection, a wall about 24 m away was used as the target to reflect the laser. The echo laser signal was collected by the receiving lens and transmitted through an AOM, in the same manner as with the modulator. The deflected laser output of the first-order diffraction fringe was focused on a PIN detector (PDA10A2 of Thorlabs) (Newton, NJ, USA). The sampling frequency was set to 20 MHz. The output of the PIN detector was given to an integrator amplifier (AD8488 of Analog Devices) (Wilmington, MA, USA) and an oscilloscope (MDO3032 of Tektronix) to monitor the signal. The integrator amplifier was controlled by the gate signal to accumulate the laser signals during the accumulation period of 1.6 µs, and the accumulated signals were sent to the computer. The combined function of the PIN detector and integrator was used to simulate the function of an integrating detector, which is normally used in range-gated lidar systems.

## 3. The Phase Code Sequence

The basic purpose of the proposed lidar system is to get remote distance measurement with high resolution. Due to the required target detection of long range, the intensity of echo signals will be low. Meanwhile, the AOM located before the detector will further reduce the intensity of echo signals. Therefore, the system should employ the pulse compression method by taking the advantage of the high SNR. This paper takes the phase-coded method as an example.

Suppose that $S_1 = [m_1, m_2, \ldots, m_N]$ represents an m-sequence with $N = 2^n - 1$ bits, where $n$ is the length of shift registers. Additionally, $m_i$, $i = 1, \ldots, N$ are the bits of the sequence. The self-correlation function can be expressed as

$$R_p(S_1, S_1) = \sum_{k=1}^{N-1-p} m_k m_{k+p} \tag{1}$$

Here, $p$ is the number of the shifted bits.

When the values of $m_i$ are only 1 and $-1$, the values of the self-correlation function become

$$R_p(S_1, S_1) = \begin{cases} N & , p = 0 \\ -1 & , p \neq 0 \end{cases} \tag{2}$$

In practical situations, the presented system will use the modulator to control the passage state of the laser signals, which will result in the modulation and demodulation having only two circumstances: pass or non-pass. The detected signal by the detector cannot be a negative value, either. Therefore, the modulation and demodulation signals can be regarded as non-negative sequences. The simplest solution is to add one to the m-sequence $S_1$, with the sequence being converted to $T_1$, whose values become 2 and 0.

$$T_1 = S_1 + 1 = [\widetilde{m}_1, \widetilde{m}_2, \ldots, \widetilde{m}_N] \tag{3}$$

The self-correlation function is expressed as

$$\begin{aligned} R_p(T_1, T_1) &= \sum_{k=1}^{N} (m_k + 1)\left(m_{k+p} + 1\right) \\ &= \sum_{k=1}^{N} \left(m_k m_{k+p} + m_k + m_{k+p} + 1\right) \end{aligned}, \tag{4}$$

For the m-sequence, the number of '1' s is one greater than the number of '0' s, which results in $\sum_{i=1}^{N} m_i = (N+1)/2$. Equation (4) becomes

$$R_p(T_1, T_1) = N + 2 + \sum_{k=1}^{N} m_k m_{k+p} \tag{5}$$

The values are $R_p(T_1, T_1) = \begin{cases} 2N + 2 & , p = 0 \\ N + 1 & , p \neq 0 \end{cases}$.

Suppose $T_1 = [\widetilde{m}_1, \widetilde{m}_2, \ldots, \widetilde{m}_N]$ is the emitted sequence of the lidar system. The period length of $T_1$ is $N$. $T_2$ is the reference sequence and $T_3$ is the echo sequence. Suppose $T_3 = [n_1, n_2, \ldots, n_N]$. Take $L$ as the phase difference of the emitted and echo sequences. The correlation function of $T_1$ and $T_3$ can be expressed as:

$$R_p(T_1, T_3) = \sum_{k=1}^{N} \widetilde{m}_k n_{k+p} = \begin{cases} 2N + 2 & , p = L \\ N + 1 & , p \neq L \end{cases} \tag{6}$$

In the proposed photonics-based lidar system, to accomplish the shift operation, the reference sequence $T_2$ is constructed based on the modulation sequence $T_1$ with the same bit width. An additional '0' is added to the end of $T_1$, where $T_2 = [\widetilde{m}_1, \widetilde{m}_2, \ldots, \widetilde{m}_N, 0]$. The period length of $T_2$ is $N + 1$. For example, if the modulation sequence $T_1$ is chosen to be a simple m-sequence [1 1 1 0 0 1 0] containing 7 bits, the reference sequence $T_2$ will be [1 1 1 0 0 1 0 0], containing 8 bits. The controlled accumulation time of the detector will be set to be the period of $T_2$. The extra bit '0' makes the correlation calculation shift automatically and the position of the highest intensity of the detector output sequence indicates the result of the correlation calculation.

Define the demodulation function of $T_2$ and $T_3$ as:

$$D_p(T_2, T_3) = \sum_{k=1}^{N+1} \widetilde{m}_k n_{k+p} = \sum_{k=1}^{N} \widetilde{m}_k n_{k+p} + \widetilde{m}_{N+1} n_{N+1+p} \tag{7}$$

According to the construction method of $T_2$, there is $\widetilde{m}_{N+1} = 0$, which leads to the result.

$$D_p(T_2, T_3) = \sum_{k=1}^{N} \widetilde{m}_k n_{k+p} = R_p(T_1, T_3) \tag{8}$$

In this way, the correlation function of $T_1$ and $T_3$ can be deduced from the demodulation function of $T_2$ and $T_3$.

To determine the position of the correlation peak during the output sequence of the detector, the modulation signal was designed with two periods, shown in Figure 3, an activation period with the system emitting the modulated laser sequences, and a silence period with no laser illumination. However, the reference sequence worked during the whole time with no silence period. The activation period contained $N + 1$ m-sequence periods. The length of the activation period equaled the length of N reference sequence periods, as the reference sequence had 1 more bit than the modulation sequence. The detector accumulated the collected energy during each reference sequence period. The length of the silence period was set according to the duty ratio requirement of the laser.

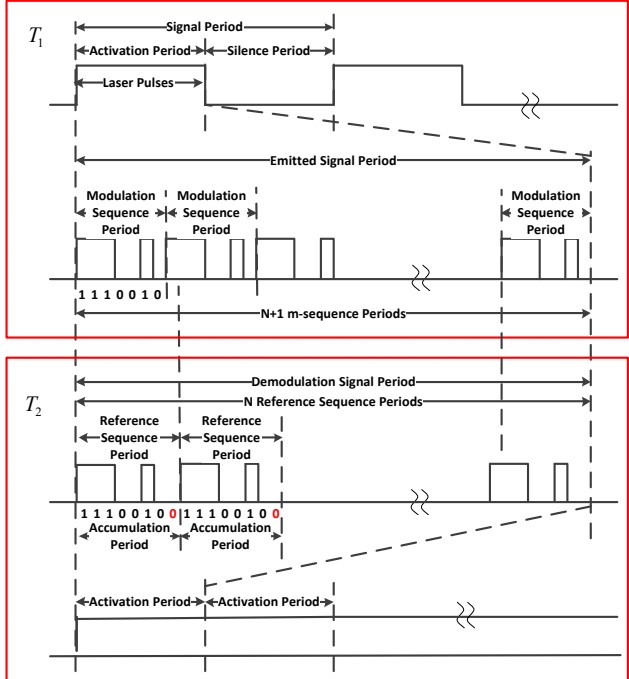

**Figure 3.** The modulation and reference sequences.

## 4. Result of the Experiment

The results of the experiment are shown in the following figures.

Figure 4a shows the real-time waveform of the PIN detector output signal monitored by the oscilloscope without accumulation. The silence period, indicated by the signals with lower values, identified the start of the correlation calculation. In the activation period, the width of each bit was 100 ns. As shown in Figure 4b, the correlation peak of the accumulated signal can be easily counted as the 3rd pulse, which indicates that the variation between the emitted laser sequence and reference sequence should be about 200 ns, which turned out to be 30 m, close to the distance of 24 m of the wall. This result demonstrates the validity of the proposed photonics-based phase-coded lidar system.

According to the result, the range resolution of the system reached 100 ns, the single bit width of transmitted signals, and the accumulation period of the PIN was 1.6 μs. This means that the sampling period of the detector is much longer than the single-bit width of the signal. For practical application, the length of the modulated sequence can be 1023 bits, which offers nanosecond resolution with a sampling rate of milliseconds. The available detectors with low performance will be capable of acquiring narrow pulse information with low sampling frequency. The cost of acquiring high-range resolution can be greatly reduced.

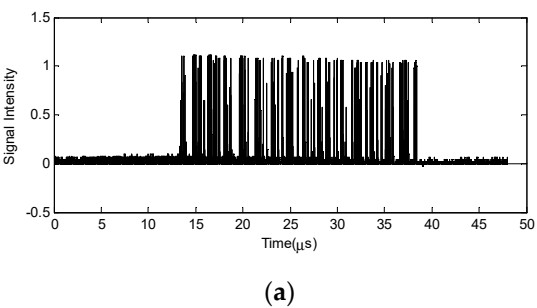

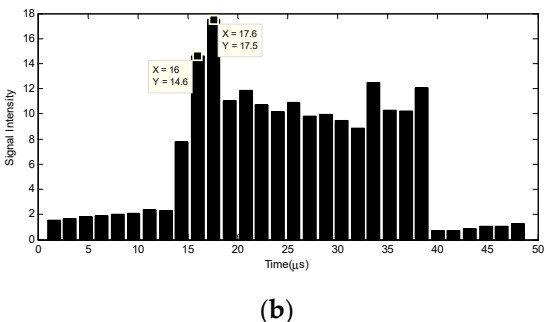

(**a**)　　　　　　　　　　　　　　　　　　　　　　　(**b**)

**Figure 4.** Signal waveforms. (**a**) Real-time waveform of the PIN output. (**b**) Accumulated signal waveform.

More importantly, as the laser has very low signal interference, the multiplication of reference signals and echo signals in laser form for thousands of pixels can be processed at the same time. This means that the imaging lidar system is capable of achieving 3D images with high-range resolution in remote detection when the detector is changed to an array detector, such as an ICCD camera, which has a very low sampling rate.

## 5. Discussion

### 5.1. Accuracy Improvement

Although the range resolution could be as narrow as the pulse width with a low sampling rate, the measurement accuracy may not be good enough yet. The pulse synchronization of the echo laser sequence and the reference sequence may cause a deviation of the correlation peak.

Suppose the echo sequence $T_3$ with length $N$ is delayed for $p + m/(m + n)$ bits. The sequence can be regarded as the combination of two separated sequences, $T_{31}$ and $T_{32}$, shown in Figure 5.

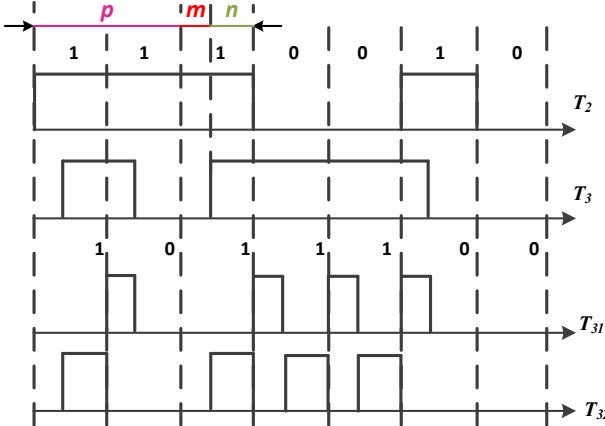

**Figure 5.** The decomposition of a delayed echo sequence.

Under the circumstance of total synchronization, $m = 0$, the correlation calculation is $R_i(T_2, T_3) = \begin{cases} N & , i = p \\ -1 & , i \neq p \end{cases}$. When $m \neq 0$, the energy of a single pulse in $T_{31}$ and $T_{32}$ becomes $m/(m + n)$ and $n/(m + n)$ of the original pulse, respectively. The correlation calculation equation shall become:

$$R_i(T_2, T_{31}) = \begin{cases} Nm/(m + n) & , i = p + 1 \\ -m/(m + n) & , i \neq p + 1 \end{cases}$$

$$R_i(T_2, T_{32}) = \begin{cases} Nn/(m + n) & , i = p \\ -n/(m + n) & , i \neq p' \end{cases}$$

(9)

Then, the correlation calculation equation shall include the addition of $R_i(T_2, T_{31})$ and $R_i(T_2, T_{32})$.

$$R_i(T_2, T_3) = \begin{cases} Nn/(m+n) - m/(m+n) & , i = p \\ Nm/(m+n) - n/(m+n) & , i = p+1 \\ -1 & , i \neq p \text{ or } p+1 \end{cases}, \quad (10)$$

If $m < n$, the peak of the correlation calculation will be at the position $p$, as $R_p(T_2, T_3) > R_{p+1}(T_2, T_3)$. If $m > n$, the peak of the correlation calculation will be at the position $p + 1$. A simulation with different delayed bits for the modulation sequence was performed. Figure 6a shows the detector output with a delay of 2 and 2/5 bits, in which case $p = 2$, $m = 2$, $n = 3$. Figure 6b shows the result of delaying 1 and 3/5 bits, in which case $p = 1$, $m = 3$, $n = 2$.

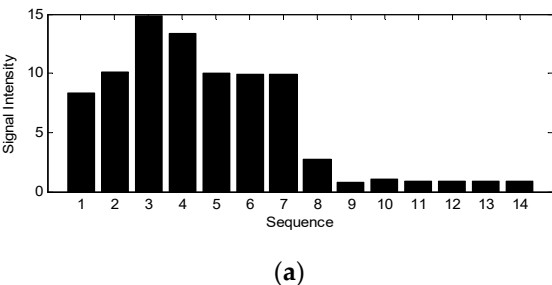

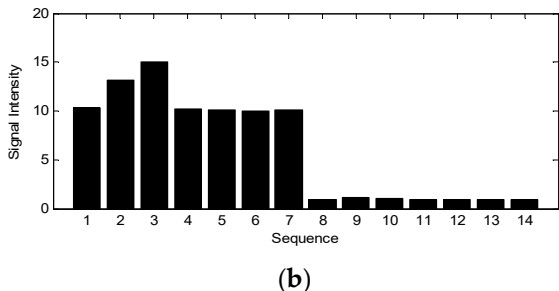

(**a**)  (**b**)

**Figure 6.** The results for varied bits delays of the echo sequence. (**a**) The result of a delay of 2 and 2/5 bits. (**b**) The result of a delay of 1 and 3/5 bits.

As shown in Figure 6a, the correlation peak remained at the 3rd pulse, representing a 2-bit delay of the echo sequence. The second-highest value of the pulses is after the correlation peak, which means that the echo sequence and the reference sequence were not synchronized and the echo sequence was delayed for more than 2 bits. The same 2-bit delay is shown in Figure 6b, as the 3rd pulse was the correlation peak. The second-highest value of the pulses is before the correlation peak, representing that the delay of the echo sequence was less than 2 bits. Therefore, the position of the second-highest pulse represents the synchronization situation of the two sequences.

The degree of synchronization can also be concluded according to the values of the peak and the second-highest value. In Figure 4b, it is very difficult to determine the values of $R_p$ and $R_{p+1}$. However, it is easy to calculate $R_p + 1$ and $R_{p+1} + 1$, which are the differences between the top two highest values and the non-peak outputs. According to Equation (9), there are $(R_p + 1) + (R_{p+1} + 1) = N + 1, R_p + 1 = (N+1)n/(m+n)$ and $R_{p+1} + 1 = (N+1)m/(m+n)$. The delay can be represented by the following equation.

$$\begin{cases} n/(m+n) = (R_p + 1)/[(R_p + 1) + (R_{p+1} + 1)] \\ m/(m+n) = (R_{p+1} + 1)/[(R_p + 1) + (R_{p+1} + 1)] \end{cases}, \quad (11)$$

As shown in Figure 4b, the peak appears at 17.6 µs with the value of 17.5, and the second-highest value appears at 16 µs with the value of 14.6, which can indicate that the phase difference between the echo signal sequence and reference sequence is less than 2 bits. Due to the noises, the values of the rest pulses are not the same. The average value of the rest pulses equals 10.5, which is used as the value of non-peak outputs. Then, $R_{p+1} + 1 = 7$ and $R_p + 1 = 4.1$, which yields the result of $p = 1$, $m/(m+n) = 0.63$. The final result becomes 24.46 m, which is very close to the real distance in the experiment.

*5.2. SNR of the System Output*

For phase-coded lidar systems, the average power of the signal of each pulse in the sequence output by the detector can be denoted as:

$$O_i = I_i f g + P_{m_i} = P_T m_i f g + P_{m_i} \tag{12}$$

where $I_i$ is the power of the emitted pulse. $P_{m_i}$ is the power of the output noise of the detector. $m_i$ is the bit of the sequence. $g$ is the responding function of the detector.

The detection process can be described as a statistical process. The power of the output signal includes the echo signal and noises. According to the research of Wu [15], the output of the detector is a Gaussian process. The noises include background noise $P_b$, dark current noise $P_d$, and thermal noise $P_{th}$. The direct detection receiver model can be simplified as:

$$p(x) = \frac{1}{\sqrt{2\pi}\sigma} \exp\left\{-\frac{(x-\mu)^2}{2\sigma^2}\right\},$$
$$\text{where } \mu = \frac{\overline{g}e}{\tau} R_L\left(\overline{k}_s + \overline{k}_b + \overline{k}_d\right), \sigma_{vo}^2 = \left(\frac{\overline{g}e}{\tau}\right)^2 F\left(\overline{k}_s + \overline{k}_b + \overline{k}_d\right) + k_{th}^2, \tag{13}$$

where $\mu$ is the mean of the output signals of the detector. $\sigma_{vo}$ is the variance of the output signals and can be regarded as the output noise. $k$ is the primary photoelectrons' counting of the photocathode. $k_s, k_b, k_d, k_{th}$ represent the photoelectrons' countings caused by echo signal, background noise, dark current noise and thermal noise. $e$ is the electron charge. $R_L$ is the load resistance. $F$ is the noise coefficient. $g$ is the system gain. $\tau$ is the sampling period.

According to Equations (12) and (13), the mean and variance of the output signal are:

$$\mu_s = P_T f g + g\left(\overline{P}_b + \overline{P}_d\right) = \overline{S}_{signal} + \overline{N}_{noise},$$
$$\sigma_C^2 = P_T f g^2 F + g^2 F\left(\overline{P}_b + \overline{P}_d\right) + \overline{P}_{th}^2, \tag{14}$$

The output SNR of a single pulse of the detector becomes:

$$SNR_{sub} = \frac{\overline{S}_{signal}}{\sigma_C} = \frac{P_T f g}{\sqrt{P_T f g^2 F + g^2 F\left(\overline{P}_b + \overline{P}_d\right) + \overline{P}_{th}^2}} \tag{15}$$

In the phase-coded system, the detector accumulates the input echo signals for the length of the sequence after the demodulator. The period of the accumulation is $(N+1)\tau$. The average power of a single bit of the echo signal can be represented as:

$$O_i = I_i f g + P_{m_i} = P_T m_i f g + P_{m_i} \tag{16}$$

The output signal after the demodulation becomes:

$$\hat{O}_i = O_i \hat{m}_{i+l} = P_T m_i \hat{m}_{i+l} f g + P_{m_i} \hat{m}_{i+l} \tag{17}$$

where $\hat{m}_{i+l}$ is the controlled bit of the reference sequence. $l = 0, 1, \ldots, n-1$ is the phase difference between the echo sequence and reference sequence.

After the accumulation in the detector, the output of the detector is:

$$\widetilde{O}_l = \sum_{i=1}^{N+1} \hat{O}_i = \sum_{i=1}^{N+1}\left(P_T m_i \hat{m}_{i+l} f g + P_{m_i} \hat{m}_{i+l}\right)$$
$$= P_T f g \sum_{i=1}^{N} m_i m_{i+l} + P_T f g m_{N+1} \hat{m}_{N+1+l} + \sum_{i=1}^{N+1} P_{m_i} \hat{m}_{i+l} \tag{18}$$

There are $\sum\limits_{i=1}^{N} m_i m_{i+l} = N$ when the phase difference $l$ is 0. According to the construction of the sequence, there is $\hat{m}_{N+1} = 0$. Therefore,

$$\widetilde{O}_{l=0} = NP_T fg + \sum_{i=1}^{N+1} P_{m_i}\hat{m}_{i+l} = NP_T fg + (N+1)\overline{P}_{noise} = S_{signal} + N_{noise} \qquad (19)$$

Then, the mean and variance of the output photons in a whole code sequence are:

$$\begin{aligned}
\mu_A &= NP_T fg + (N+1)g\left(\overline{P}_b + \overline{P}_d\right) = \overline{S}_{signal} + \overline{N}_{noise}, \\
\sigma_{\widetilde{C}}^2 &= NP_T fg^2 F + (N+1)\left[g^2 F\left(\overline{P}_b + \overline{P}_d\right) + \overline{P}_{th}^2\right]
\end{aligned} \qquad (20)$$

The SNR of the output of the lidar system becomes:

$$\begin{aligned}
SNR_A &= \frac{S_{signal}}{\sigma_{\widetilde{C}}} = \frac{NP_T fg}{\sqrt{NP_T fg^2 F + (N+1)\left[g^2 F\left(\overline{P}_b + \overline{P}_d\right) + \overline{P}_{th}^2\right]}}, \\
&\approx \sqrt{N}\frac{P_T fg}{\sqrt{P_T fg^2 F + g^2 F\left(\overline{P}_b + \overline{P}_d\right) + \overline{P}_{th}^2}} = \sqrt{N}SNR_{sub}
\end{aligned} \qquad (21)$$

According to Equation (21), the SNR of the lidar system is improved by $\sqrt{N}$ times the SNR of a single bit of the sequence.

## 6. Conclusions

In summary, a photonics-based phase-coded lidar system is proposed in this paper. The system includes two modulators for modulation and demodulation, respectively. The modulator, such as an AOM, modulates the laser with constant intensity into an encoded pulse sequence. The demodulator is located between the collection antenna and the detector, and its function is to multiply the electrical reference sequence with the laser echo sequence. The detector, such as a PIN or an ICCD, with the gate controller, integrates the output of the demodulator at a very low speed. The combination of the demodulator and detector accomplishes the correlation calculation of the echo laser sequence in the form of laser and reference sequences in the form of electrical signals.

The system employs an m-sequence as the modulation sequence. A reference sequence is constructed based on the modulation sequence with an additional '0' bit. The accumulation period of the detector is set to the period of the reference sequence. The output of the detector demonstrates the delayed phase of the laser echo sequence compared to the reference sequence.

This photonics-based signal processing method has transferred the classical correlation calculation electrically into an optical process. Because the laser beams have low signal interference, the system is capable of processing multiple pixels at the same time, resulting in great signal bandwidth expansion. Moreover, the system may achieve 3D imaging at higher range resolution, with the available devices, at low sampling rates. This potential implementation may have great impacts on imaging lidar systems with array detectors at low sampling rates and limited signal bandwidth.

**Author Contributions:** Conceptualization, formal analysis, investigation, writing, S.C. and L.W.; methodology, reviewing, L.X.; validation, Y.Z. and J.Z. All authors have read and agreed to the published version of the manuscript.

**Funding:** This work was supported in part by the National Natural Science Foundation of China (NSF) under Grant 61801429. This work was also supported in part by the Natural Science Foundation of Zhejiang Province under Grant LY20F010001 and LQ20F050010.

**Institutional Review Board Statement:** Not applicable.

**Informed Consent Statement:** Not applicable.

**Data Availability Statement:** Not applicable.

**Conflicts of Interest:** The authors declare no conflict of interest.

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
