# Peer review of "Photonic Signal Processing in Phase-Coded Lidar System"

_photonics, doi:10.3390/photonics10050598_

Round 1

Reviewer 1 Report

The manuscript is focused on the phased coded lidar systems. In particular the system adopts a photonic processing technique to achieve high precision measurements and long distances with low requirements for the sampling electronics at the receiver side. However, there should be some issues to be clarified.

1.Resolution and SNR.

The method has an inherent tradeoff between range resolution and SNR, as temporal encoding using AOM reduce the weak echo signals. Moreover, as the distance is decoded from the correlation results, a poor SNR seems to a larger effect on the range measurement as compared to the ordinary pulsed-ToF method (e.g. the reconstruction algorithm in V is based on the correlation amplitudes). To clearly address those concerns, I would recommend the author to include the SNR calculation of the system.

2.The shape of sub-pulses

The considered pulses are with the rectangle shape. But the actual shape of the pulses should be considered as a Gaussian distributed pulse. Does the shape of the pulses affect the expected results?

3.Range resolution

The paper shows the width of the sub-pulses is 100ns. The range resolution seems a little low. Why do the authors choose sub parameter for the width of pulses? Could you choose other hardware to improve the performance?

4.Ranging accuracy

To improve the ranging accuracy, the manuscript will calculate the ratio between the accumulated neighboring outputs. If the SNR is low, how to choose the required outputs?

5.Range imaging

If the lidar is going to get 3D range images, the system may choose ICCD as the detectors. But ICCD has a very low sampling speed. Then how to decide the coding pattern to balance the range resolution and low sampling speed?

6.Misunderstanding subscript number

In line 143 in page 4,   is the reference sequence and   is the echo sequence. But in line 227 in page 7,   becomes the echo sequence. Could the authors clarify the meaning of the sequences?

7.Wrong font

In line 227 in page 7, the length of the sequence is N. This variable should be in Italic.

Author Response

Comments:

The manuscript is focused on the phased coded lidar systems. In particular the system adopts a photonic processing technique to achieve high precision measurements and long distances with low requirements for the sampling electronics at the receiver side. However, there should be some issues to be clarified.

  1. Resolution and SNR.

The method has an inherent tradeoff between range resolution and SNR, as temporal encoding using AOM reduce the weak echo signals. Moreover, as the distance is decoded from the correlation results, a poor SNR seems to a larger effect on the range measurement as compared to the ordinary pulsed-ToF method (e.g. the reconstruction algorithm in V is based on the correlation amplitudes). To clearly address those concerns, I would recommend the author to include the SNR calculation of the system.

Response 1: Thank the reviewer for the comment. The answers are as follows:

We have added Section 5.2 to calculate the SNR of the system. The analysis shows that the phase coded system will improve the SNR for  times compared with a single pulse in the sequence.

Author action: We have updated the manuscript as described above.

In the revised manuscript:

The changed parts are in Section 5.2.

All the changes are marked in red words.

The changed part:

5.2 SNR of the system output

For phase coded lidar system, the average power of the signal of each pulse in the sequence output by the detector can be denoted as:

,

(12)

where  is the power of the emitted pulse.  is the power of the output noise of the detector .  is the bit of the sequence.  is the responding function of the detector.

The detection process can be described as a statistical process. The power of the output signal includes the echo signal and noises. According to the research of Wu[15], The output of the detector is a Gaussian processes. The noises include background noise , dark current noise , and thermal noise . The direct detection receiver model can be simplified as:

,

(13)

where ,

where  is the mean of the output signals of the detector.  is the variance of the output signals and can be regarded as the output noise.  is the primary photoelectrons counting of the photocathode., , , represent the photoelectrons countings caused by echo signal, background noise, dark current noise and thermal noise.  is the electron charge.  is the load resistance.  is the noise coefficient. is the system gain.  is the sampling period.

According equation (12) and (13), the mean and variance of the output signal are:

,

(14)

,

The output SNR of a single pulse of the detector becomes:

,

(15)

In the phase coded system, the detector accumulates the input echo signals for the length of the sequence after the demodulator. The period of the accumulation is . The average power of a single bit of the echo signal can be represented as:

,

(16)

The output signal after the demodulation becomes:

,

(17)

where is the controlled bit of the reference sequence.is the phase difference between the echo sequence and reference sequence.

After the accumulation in the detector, the output of the detector is:

,

(18)

There are , when the phase difference is 0. According to the construction of the sequence, there is . Therefore,

,

(19)

Then the mean and variance of the output photons in a whole code sequence are:

,

(20)

,

The SNR of the output of the lidar system becomes:

,

(21)

According to equation (21), the SNR of the lidar system is improved by  times of the SNR of a single bit of the sequence.

  1. The shape of sub-pulses

The considered pulses are with the rectangle shape. But the actual shape of the pulses should be considered as a Gaussian distributed pulse. Does the shape of the pulses affect the expected results?

Response 1: Thank the reviewer for the comment. The answers are as follows:

The theoretical analysis in the manuscript is indeed based on rectangle pulses, because it is simple to analyze the effect of the encoding algorithm. Each bit of the sequence is then regarded as the simplest number of 1 or 0. And the analysis shows that it will have good performance for such system.

The actual shape of the pulses is actually Gaussian distributed pulse. For the proposed algorithm, the detector is controlled to accumulate the energy of all the demodulated pulses. This means all the energy of the pulse, no matter of the shape of the pulses, will be added together. Only the pulses at the beginning and ending of the sequence will be affected by the shape of the pulses due to the rising and falling edge of the pulses. But considering the length of the sequence, which normally includes more than 1000 bits, the errors caused by the shape of the pulses will be minimized.

The algorithm to improve the accuracy may be affected by the shape of the pulses, because the calculation of the accurate position depends on the degree of synchronization. As the different degree of synchronization cause the variation of the proportion of the 2 sub-pulses in 1 bit, the calculation may lower the accuracy of the measurement. But because the sub-pulses are accumulated, only the first and last sub-pulses will be affected by the Gaussian distributed shape of the pulses.

Therefore, the actual Gaussian distributed shape of the pulses will do little harm to the accuracy of the measurement.

3.Range resolution

The paper shows the width of the sub-pulses is 100ns. The range resolution seems a little low. Why do the authors choose such parameter for the width of pulses? Could you choose other hardware to improve the performance?

Response 1: Thank the reviewer for the comment. The answers are as follows:

We had several reasons to choose the width of the sub-pulses to be 100ns. First of all, the laser needs to be modulated to multiple pulses. The high performance of the modulator is required. But unfortunately, the available modulator in the lab can reach high modulation frequency constantly and stably. This makes us to make a choice to balance the parameters. Secondly, we wanted to simulate the accuracy improvement. Therefore we decided to use the detector with high speed to give multiple outputs during the pulse. More accurate as considered it is more outputs are given. Therefore, we chose 100ns as the width of the sub-pulses to contain almost 97 samples for a single bit in the experiment. Then we had adequate samples to be calculated to remove the errors caused by detectors. Furthermore, in such width of the pulses, the experiment can still achieve the sub-millimeter scale accuracy.

It is only in the experiment that we chose the width of the sub-pulses as 100ns. In actual application of the lidar system, the system may have better choices for both the modulators and detectors for narrow pulses and faster sampling frequency. The actual accuracy can be improved and better than the millimeter scale accuracy.

4.Ranging accuracy

To improve the ranging accuracy, the manuscript will calculate the ratio between the accumulated neighboring outputs. If the SNR is low, how to choose the required outputs?

Response 1: Thank the reviewer for the comment. The answers are as follows:

Due to the design of the accuracy improvement algorithm, the peak of the correlation calculation of the reference and echo sequences is split into 2 pulses. According to the added SNR analysis section, the SNR of the lidar system is improved by  times of the SNR of a single bit of the sequence. This means that the peak of the correlation calculation is higher than other output pulses. If the SNR is low, the system may increase the length of the sequence which will achieve higher SNR. Otherwise, the system must find the peak wisely, such as that the sum of the adjacent output pulses should be the highest. If the neighbouring outputs are not showing the same pattern, having high intensity of the pulse, the chosen pulse may be not the peak of the correlation calculation.

5.Range imaging

If the lidar is going to get 3D range images, the system may choose ICCD as the detectors. But ICCD has a very low sampling speed. Then how to decide the coding pattern to balance the range resolution and low sampling speed?

Response 1: Thank the reviewer for the comment. The answers are as follows:

As the reviewer mentioned, the ICCD detector has an extreme low sampling speed. In this case, the system should first decide the detection range and the resolution request of the scene, which will impact the length of the sequence. At the same time the detection framerate will be limited by the square of the length of the sequence. If the length of the sequence and the resolution request of the scene have been determined and the framerate of the detection can be satisfied, the length of the silence period can be decided according to other requests.

6.Misunderstanding subscript number

In line 143 in page 4,  is the reference sequence and  is the echo sequence. But in line 227 in page 7,  becomes the echo sequence. Could the authors clarify the meaning of the sequences?

Response 1: Thank the reviewer for the comment. The answers are as follows:

The reviewer is correct. We had made mistakes. Now we have corrected the words.  is the reference sequence and  is the echo sequence.

Author action: We have updated the manuscript as described above.

In the revised manuscript:

The changed parts are from line 227 to 237.

All the changes are marked in red words.

The changed part:

Although the range resolution can be as narrow as the pulse width with a low sampling rate, the measurement accuracy may not be good enough yet. The pulse synchronization of the echo laser sequence and the reference sequence may cause the deviation of the correlation peak.

Suppose the echo sequence with length N is delayed for  bits. The sequence can be regarded as the combination of 2 separated sequences  and , shown in Figure 5.

Figure 5. The decomposition of delayed echo sequence

Under the circumstance of total synchronization, , the correlation calculation is . When , the energy of a single pulse in  and  becomes and  of the original pulse respectively. The correlation calculation equation shall become:

,

(9)

,

Then, the correlation calculation equation shall be the addition of  and .

,

(10)

7.Wrong font

In line 227 in page 7, the length of the sequence is N. This variable should be in Italic.

Response 1: Thank the reviewer for the comment. The answers are as follows:

The reviewer is correct. We had made mistakes. Now we have corrected the words.

Author action: We have updated the manuscript as described above.

In the revised manuscript:

The changed parts are in line 164, 227.

All the changes are marked in red words.

The changed part:

In line 164: The activation period contains N+1 m-sequence periods.

In line 227: Suppose the echo sequence with length N is delayed for  bits.

Thank you again for your suggestions. We have already made the modifications. We hope this will meet your demands.

Have a nice day.

Reviewer 2 Report

The comments are attached below for your reference.

The English writing should be checked and improved. 

Author Response

Comments:

Time-of-flight method is widely used in ranging and 3D imaging applications for its simple setup, high SNR, and a decent resolution. As the resolution is proportional to pulse width, a high bandwidth Tx/Rx system is required. The author proposed to use temporal encoded pulse (using AOM) and correlation (using PD) to mitigate the requirement on the bandwidth requirement, which is an interesting technique for signal processing. However, there may be some inherent concerns with the idea:

  1. CW modulated ToF method is widely used in ToF cameras (e.g. ESPROS ToF 611 camera) which determines the distance by calculating the correlation between illumination and echo signals that is usually sinusoidally modulated. Instead of sampling the whole waveform, it samples the signal at a few fixed temporal windows and calculates the phase of the modulation by comparing the measured amplitude, for example, using the 4-bucket algorithm. By using high-resolution ADC, it can achieve 0.1mm at 10MHz modulation rate, which is much better than the accuracy presented by the proposed method. I think a detailed comparison with the CW-ToF method in the introduction or discuss session is necessary to give readers a better understanding of the pros/cons of the proposed idea. The author could refer to the well-oriented tutorial on the CW-ToF method from the lecture notes here: http://perception.inrialpes.fr/people/Horaud/Courses/pdf/Horaud_3DS_3.pdf

Response 1: Thank the reviewer for the comment. The answers are as follows:

The reviewer is correct. The CW-ToF method does have more accurate measurement. The detailed comparison is made as follows:

The full name of ToF is Time of Flight. It belongs to LiDAR technology and is also one of the main 3D recognition technologies at present.

ToF is divided into two types: dToF (direct) and iToF (indirect). Among them, dToF, the full name is direct Time of Flight. As the name suggests, dToF directly measures flight time. The principle is to directly use nanosecond or even picosecond short pulse lasers at the emission end, and to react quickly after emission, quickly receiving the reflected laser. Therefore, a higher level of progress is required for single photon Avalanche Diode (SPAD) or APD (avalanche photodiode) detection.

DToF calculates the distance by recording the time interval between the transmitted and received pulses. DToF will transmit and receive N optical signals within a single frame measurement time, and then analyze and calculate these times to obtain the final distance.

IToF, as the name suggests, uses indirect time of flight to reveal distance judgments. The principle is to modulate the emitted light wave into a periodic signal of a certain frequency which is normally a CW(Continuous Wave) signal , and indirectly calculate the flight time of the light by measuring the phase difference between the transmitted signal and the signal reflected back by the measured object to reach the receiving end, thereby obtaining depth data. Due to its different principles, it can use multiple periodic models to determine phase difference. In practical use, due to the use of periodic frequency wave strategy, in order to better ensure accuracy, it is necessary to receive higher frequencies and better results. Therefore, the farther the distance, the more difficult it is to distinguish the wavelengths of the two cycles, which may cause interference. Therefore, the application distance is shorter. And compared to dToF, it is prone to strong light interference.

In terms of cost, due to the need for high-precision transmitting and receiving equipment, as well as the need for time synchronization judgment between the two pulses, dToF will incur a certain amount of cost compared to iToF in practical use. However, the cost of iToF is lower and it can be used on more devices to achieve distance estimation. It can also be said that it is currently the first choice for many devices equipped with ToF.

In terms of accuracy, dToF cannot achieve high image resolution, while iToF requires a larger sensor size to receive wavelength changes of light due to differences in its principles. Therefore, it can achieve higher image resolution compared to dToF in object recognition, 3D reconstruction, and other aspects.

In terms of power consumption, the pulse wave emission used by dToF has a lower duty cycle compared to the continuous wave emission of iToF. The system can emit more targeted light sources at the same time. Compared to the two measurements, dToF has lower power consumption and is more suitable for use on devices with lower power consumption.

Author action: We have updated the manuscript as described above.

In the revised manuscript:

The changed parts are in line 53.

All the changes are marked in red words.

The changed part:

The approach of time of flight (ToF) measurement is considered as a promising technique[25]. The approach known as direct ToF (dToF) calculates the time directly from an accurate time base[26, 27]. It is generally considered a low-resolution technology, requiring complex (and expensive) mechanical scanning to achieve high resolutions. The approach of indirect ToF (iToF) calculates distance based on a phase shift to a known reference signal. This technique suits the high-resolution applications with less detection range[28]. But iToF technique may extend the ranging depth to avoid the ambiguity of phase by combining the detection of double frequencies. The approach makes iToF vary useful in forward-looking applications where other technologies have limitations.

  1. I'm curious how different coding algorithm could affect the performance of the proposed idea. In phase-coded radar, a special class of binary codes (e.g. Barker codes) is the optimum. Could the author talk about what would be the criteria for choosing the binary coding algorithm?

Response 1: Thank the reviewer for the comment. The answers are as follows:

Phase encoding ranging is mainly achieved by measuring the phase of the echo signal compared with the reference signal. The encoding algorithms require that a higher peak of the encoded autocorrelation function. For a certain length of codes, its autocorrelation function should have the smallest peak sidelobe. These codes have low sidelobe of autocorrelation functions or zero Doppler response, which are exactly what pulse compression radar requires. Both Buck codes and combined Buck codes have this characteristic. But the length of the Barker code is too small, which limits its practical application. Therefore the combined Barker code is widely used.

  1. The considered sub-pulses seem to be a regulated pulse shape. The noises are not considered much. How to deal with the signals with low SNR?

Response 1: Thank the reviewer for the comment. The answers are as follows:

Due to the design of the accuracy improvement algorithm, the peak of the correlation calculation of the reference and echo sequences is split into 2 pulses. According to the added SNR analysis section, the SNR of the lidar system is improved by  times of the SNR of a single bit of the sequence. This means that the peak of the correlation calculation is higher than other output pulses. If the SNR is low, the system may increase the length of the sequence which will achieve higher SNR. Otherwise, the system must find the peak wisely, such as that the sum of the adjacent output pulses should be the highest. If the neighbouring outputs are not showing the same pattern, having high intensity of the pulse, the chosen pulse may be not the peak of the correlation calculation.

  1. In line 143, is referred as the reference sequence and is referred as the echo sequence. But in line 227,  becomes the echo sequence. The authors should stick to the same the meaning of variables and subscripts.

Response 1: Thank the reviewer for the comment. The answers are as follows:

The reviewer is correct. We had made mistakes. Now we have corrected the words.  is the reference sequence and  is the echo sequence.

Author action: We have updated the manuscript as described above.

In the revised manuscript:

The changed parts are from line 227 to 237.

All the changes are marked in red words.

The changed part:

Although the range resolution can be as narrow as the pulse width with a low sampling rate, the measurement accuracy may not be good enough yet. The pulse synchronization of the echo laser sequence and the reference sequence may cause the deviation of the correlation peak.

Suppose the echo sequence with length N is delayed for  bits. The sequence can be regarded as the combination of 2 separated sequences  and , shown in Figure 5.

Figure 5. The decomposition of delayed echo sequence

Under the circumstance of total synchronization, , the correlation calculation is . When , the energy of a single pulse in  and  becomes and  of the original pulse respectively. The correlation calculation equation shall become:

,

(9)

,

Then, the correlation calculation equation shall be the addition of  and .

,

(10)

  1. In Figure 5, is divided into 2 sub-sequences, and . Why does  get the first part, not the 2nd part? Will the selection affect the final result of the detection?

Response 1: Thank the reviewer for the comment. The answers are as follows:

There is no difference for the name of the 2 sub-pulses. Because of the problem of pulse synchronization of the echo laser sequence, the pulses are divided into 2 parts. The improvement algorithm is trying to applying the ratio calculation of the 2 sub-pulse into the estimation of the accurate measurement of the targets.

Therefore we had such equations to do the calculation:

  1. If the target is far from the lidar system, the length of the coding of the sequence must be huge according to the manuscript. In this case, the time for one frame of the detection must be even longer, which can not achieve the detection in real time. Then how to make the tradeoff between the detection range and the frametime?

Response 1: Thank the reviewer for the comment. The answers are as follows:

As the reviewer noticed, the length of the coding sequence should be long enough to make sure the detection range is long enough. But on another hand, the long sequence will result in the long detection period and low detection frame. These are conflict parameters.

To solve this problem, the designer of the lidar system may break the long sequence into 2 sequences with different lengths. Each short sequence will have a short detection range. The detection range will be regarded as a separated ranging unit. The total range of the target will be broken into several unit with a certain phase of each unit. In each time, when a short sequence is used, the lidar system will return the number which indicates the phase. If 2 short sequences are used, the final detection range will equal to the least common multiple of the 2 short units. In actual application of the lidar system, 2 or 3 short sequences are used for the detection of long range targets.

For example, if the target is 600 meters away, the system employs 2 short sequences. The 1st sequence has a detection range of 70 m. The 2nd sequences has a detection range of 90. In the detection with the 1st sequence, the reading should be 40 m because the lidar system will not tell the operator how many cycles the signal has experience. The lidar system can not detect the phase. Therefore 600=8×70+40. Therefore the final reading for the detection of the 1st sequence is 40m. The same calculation will be done for the detection of the 2nd sequence. 600=6×90+60. Therefore the 2nd reading will be 60m. Through calculation, the final result will be 600m. And the detection range for the combined 2 sequences, will be 630m. If 2 sequences are not enough, the system may employ more sequences.

The advantage of short sequences is that the system has faster frame rate. The period of the detection for short sequence will be short. The total detection period increased linearly with the number of the sequences, not the exponential growth. It saves time.

  1. The English writing can be double checked and improved. For example, in line 93, the sentence of “The laser with constant intensity is modulated.” should be “The lase with THE constant intensity is modulated.”.

Response 1: Thank the reviewer for the comment. The answers are as follows:

The reviewer is correct. We had made mistakes. Now we have corrected the words.

Author action: We have updated the manuscript as described above.

In the revised manuscript:

The changed parts are in line 93.

All the changes are marked in red words.

The changed part:

The laser with the constant intensity is modulated by the modulator into a phase coded sequence  and transmits through the transmission antenna illuminating the target.

Thank you again for your suggestions. We have already made the modifications. We hope this will meet your demands.

Have a nice day.

Reviewer 3 Report

This paper describes a practical methodology with phase coded lidar system. The purpose, methodology and investigation result are all good and would attract peoples' interest those who are involved in this field of research works. There are couple of items those would provide clearer description for this paper. One is a structure of the description. The manuscript consists of introduction, the photonics signal processing in lidar system, the phase code sequence, result of the experiment, accuracy improvement, and conclusion. Actually, the first part of result of the experiment is description of experiments and such portion should be a part of experimental section. The other is related to the structure. There are many relevant previous works such as "J. Zmnicki, Spatial heterodyne imaging using a broadband source, M.S. thesis, University of Dayton 2018". Current introduction looks too narrow to discuss this investigation, and due to small view, current structure does not represent the scope of the work in adequate manner. It would be improved to describe wider scope of introduction including proper citations, and phase coding and experimental section would be combined to clarify the scope of the work. The accuracy improvement would be a part of discussion. At the discussion section, some diffraction limit discussion would be necessary to verify this method's superiority. There are some technical clarifications would be necessary. At the last portion of result of the experiment (page 7), how to adjust each pixel's in-coming signal intensity to keep the high resolution image? It would be required what exact measurement condition was applied for detector's control. Also, it may need to make some allusion on a possible image quality improvement opportunity such as each pixel's in-coming signal comparison and weighting amplifying algorithm for high resolution image. 

Overall description is good.

Author Response

Comments:

This paper describes a practical methodology with phase coded lidar system. The purpose, methodology and investigation result are all good and would attract peoples' interest those who are involved in this field of research works. There are couple of items those would provide clearer description for this paper.

  1. One is a structure of the description. The manuscript consists of introduction, the photonics signal processing in lidar system, the phase code sequence, result of the experiment, accuracy improvement, and conclusion. Actually, the first part of result of the experiment is description of experiments and such portion should be a part of experimental section.

Response 1: Thank the reviewer for the comment. The answers are as follows:

The comment is very necessary. We have moved the description of the experiment to Section 2 following the description of the system structure. The parameters of the setup of the experiment will give the readers an exclusive understanding of the system.

Author action: We have updated the manuscript as described above.

In the revised manuscript:

Section 4.1 is removed. And the descriptions of the experiment setup are added to Section 2.

All the changes are marked in red words.

The changed part:

As shown in Figure 2, a demonstration experiment is proposed to validate of the proposed photonics-based phase coded lidar system.

Figure 2. The demonstration experience setup

The system employs a CW laser transmitter emitting a laser signal with constant intensity, at the 532nm wavelength with the output power 75mW. The laser is modulated to the coded pulse sequence by the acousto-optical modulator (AOM RF200 of Qingjin). The signal generator (AFG3102C of Tektronix) generates two synchronous signals, the modulation sequence and reference sequence. The modulation sequence is selected as an m-sequence of 15 bits long with the bit width of 100ns. Therefore, the period of the modulation and reference sequences are 1.5 with 15 bits and 1.6with 16 bits.  To simplify the experiment, the activation period and silence period are both set to 24.

Due to the Prague diffraction effect of the AOM, the direction of the modulated laser signal is deflected to the first-order diffraction fringe. As the intensity of the laser signal is not high enough to support remote target detection, a wall about 24 meters away is used as the target to reflect the laser. The echo laser signal is collected by the receiving lens and transmits through an AOM, the same as the modulator. The deflected laser output of the first-order diffraction fringe is focused on a PIN detector (PDA10A2 of Thorlabs). The sampling frequency is set to 20MHz. The output of the PIN detector is given to an integrator amplifier (AD8488 of Analog Devices) and an oscilloscope (MDO3032 of Tektronix) to monitor the signal. The integrator amplifier is controlled by the gate signal to accumulate the laser signals during the accumulation period of 1.6, and the accumulated signals are sent to computer. The combined function of the PIN detector and integrator is used to simulate the function of an integrating detector, normally used in range gated lidar system.

  1. The other is related to the structure. There are many relevant previous works such as "J. Zmnicki, Spatial heterodyne imaging using a broadband source, M.S. thesis, University of Dayton 2018". Current introduction looks too narrow to discuss this investigation, and due to small view, current structure does not represent the scope of the work in adequate manner. It would be improved to describe wider scope of introduction including proper citations, and phase coding and experimental section would be combined to clarify the scope of the work.

Response 1: Thank the reviewer for the comment. The answers are as follows:

The reviewer is correct. We had added more reference to compare the dToF and iToF techniques.

Author action: We have updated the manuscript as described above.

In the revised manuscript:

The changed parts are in line 53.

All the changes are marked in red words.

The changed part:

The approach of time of flight (ToF) measurement is considered as a promising technique[25]. The approach known as direct ToF (dToF) calculates the time directly from an accurate time base[26, 27]. It is generally considered a low-resolution technology, requiring complex (and expensive) mechanical scanning to achieve high resolutions. The approach of indirect ToF (iToF) calculates distance based on a phase shift to a known reference signal. This technique suits the high-resolution applications with less detection range[28]. But iToF technique may extend the ranging depth to avoid the ambiguity of phase by combining the detection of double frequencies. The approach makes iToF vary useful in forward-looking applications where other technologies have limitations.

  1. The accuracy improvement would be a part of discussion. At the discussion section, some diffraction limit discussion would be necessary to verify this method's superiority. There are some technical clarifications would be necessary.

Response 1: Thank the reviewer for the comment. The answers are as follows:

We have noticed that the discussion part is not enough. We have added Section 5.2 to calculate the SNR of the system. The analysis shows that the phase coded system will improve the SNR for  times compared with a single pulse in the sequence.

Author action: We have updated the manuscript as described above.

In the revised manuscript:

The changed parts are in Section 5.2.

All the changes are marked in red words.

The changed part:

5.2 SNR of the system output

For phase coded lidar system, the average power of the signal of each pulse in the sequence output by the detector can be denoted as:

,

(12)

where  is the power of the emitted pulse.  is the power of the output noise of the detector .  is the bit of the sequence.  is the responding function of the detector.

The detection process can be described as a statistical process. The power of the output signal includes the echo signal and noises. According to the research of Wu[15], The output of the detector is a Gaussian processes. The noises include background noise , dark current noise , and thermal noise . The direct detection receiver model can be simplified as:

,

(13)

where ,

where  is the mean of the output signals of the detector.  is the variance of the output signals and can be regarded as the output noise.  is the primary photoelectrons counting of the photocathode., , , represent the photoelectrons countings caused by echo signal, background noise, dark current noise and thermal noise.  is the electron charge.  is the load resistance.  is the noise coefficient. is the system gain.  is the sampling period.

According equation (12) and (13), the mean and variance of the output signal are:

,

(14)

,

The output SNR of a single pulse of the detector becomes:

,

(15)

In the phase coded system, the detector accumulates the input echo signals for the length of the sequence after the demodulator. The period of the accumulation is . The average power of a single bit of the echo signal can be represented as:

,

(16)

The output signal after the demodulation becomes:

,

(17)

where is the controlled bit of the reference sequence.is the phase difference between the echo sequence and reference sequence.

After the accumulation in the detector, the output of the detector is:

,

(18)

There are , when the phase difference is 0. According to the construction of the sequence, there is . Therefore,

,

(19)

Then the mean and variance of the output photons in a whole code sequence are:

,

(20)

,

The SNR of the output of the lidar system becomes:

,

(21)

According to equation (21), the SNR of the lidar system is improved by  times of the SNR of a single bit of the sequence.

  1. At the last portion of result of the experiment (page 7), how to adjust each pixel's in-coming signal intensity to keep the high resolution image? It would be required what exact measurement condition was applied for detector's control. Also, it may need to make some allusion on a possible image quality improvement opportunity such as each pixel's in-coming signal comparison and weighting amplifying algorithm for high resolution image.

Response 1: Thank the reviewer for the comment. The answers are as follows:

The manuscript did mention that the proposed algorithm may leading to possible imaging improvements. As we all know the spatial light modulation has the disadvantage of long modulation period, which results in bad ranging resolution. This is because the control for each pixel is different to each other. That increase the control period. The controlled mechanism also affects the speed of modulation. For example, the liquid crystal spatial light modulator may reach maximum 1kHz.

But in the proposed algorithm, the control for all signal demodulation should be the same. In the design of an adapted optics with EOM in the study of Schael[1], the echo signals from all the space are directed into an adapted telescope and illuminate the camera. The EOM is of the transverse field type. EOM’s are installed on the side of the transmitter. Along focal length lens is used to reduce the laser beam diameter without increasing the divergence. However the beam must enter and exit the modulator with little distortion.

In Schael’s design, the control for all the pixels is achieved. In this case, although the camera with multiple pixels has low sampling rates, the system can still achieve high ranging resolution and accuracy.

Figure 1 Adapted optics with EOM

[1] Schael U, Rothe H. Field measurements with 1574-nm imaging and scannerless eye-safe laser radar[C]. Proc. of SPIE: Laser Radar Technology and Applications VI, 4377, 1-11, (2001).

Thank you again for your suggestions. We have already made the modifications. We hope this will meet your demands.

Have a nice day.

Round 2

Reviewer 3 Report

Most of my suggested comments at the original manuscript were revised in favorite manner. A potential improvement would be a little further comparison with previous works to clarify superiority of this approach, though.

Author Response

Point 1:  Most of my suggested comments at the original manuscript were revised in favorite manner. A potential improvement would be a little further comparison with previous works to clarify superiority of this approach, though.

Response 1: Thank the reviewer for the comment. The answers are as follows:

We have added some references about the phase coded sequence applications in lidar system in the introduction section although we could not find the M.S. thesis of "J. Zmnicki, Spatial heterodyne imaging using a broadband source". We hope this will meet the requirement of the reviewer.

Author action: We have updated the manuscript as described above.

In the revised manuscript:

The changed part is in line 63.

All the changes are marked using the “Track Changes” function.

The changed part:

Yang built a phase-coded modulation lidar based on ghost imaging algorithm to get 3D images of distant targets in 2020[30]. Ding presented a Coded-pulse-bunch-laser-based single-photon lidar for fast long-distance ranging in 2022[31].

Thank you again for your suggestions.
